# Senescence-Associated Heterochromatin Foci Suppress γ-H2AX Focus Formation Induced by Radiation Exposure

**DOI:** 10.3390/ijms25063355

**Published:** 2024-03-15

**Authors:** Takashi Oizumi, Tomoya Suzuki, Junya Kobayashi, Asako J. Nakamura

**Affiliations:** 1Department of Biological Sciences, College of Science, Ibaraki University, Mito 310-8512, Japan; takashi.oizumi.jc11@vc.ibaraki.ac.jp (T.O.); 23nd104g@vc.ibaraki.ac.jp (T.S.); 2School of Health Sciences at Narita, International University of Health and Welfare, Chiba 286-8686, Japan; kobayashij@iuhw.ac.jp

**Keywords:** γ-H2AX, DSB repair, heterochromatin, cellular senescence

## Abstract

DNA damage is induced by both endogenous and exogenous factors. Repair of DNA double-strand break (DSB), a serious damage that threatens genome stability, decreases with senescence. However, the molecular mechanisms underlying the decline in DNA repair capacity during senescence remain unclear. We performed immunofluorescence staining for phosphorylated histone H2AX (γ-H2AX) in normal human fetal lung fibroblasts and human skin fibroblasts of different ages after chronic irradiation (total dose, 1 Gy; dose rate, 1 Gy/day) to investigate the effect of cellular senescence and organismal aging on DSB repair. Accumulation of DSBs was observed with cellular senescence and organismal aging, probably caused by delayed DSB repair. Importantly, the formation of γ-H2AX foci, an early event in DSB repair, is delayed with cellular senescence and organismal aging. These results suggest that the delay in γ-H2AX focus formation might delay the overall DSB repair. Interestingly, immediate γ-H2AX foci formation was suppressed in cells with senescence-associated heterochromatin foci (SAHF). To investigate the relationship between the γ-H2AX focus formation and SAHF, we used LiCl to relax the SAHFs, followed by irradiation. We demonstrated that LiCl rescued the delayed γ-H2AX foci formation associated with cellular senescence. This indicates that SAHF interferes with γ-H2AX focus formation and inhibits DSB repair in radiation-induced DSB. Our results suggest that therapeutic targeting of SAHFs have potential to resolve DSB repair dysfunction associated with cellular senescence.

## 1. Introduction

DNA damage is induced by factors such as radiation, chemicals, and replication stress. DNA damage, especially DNA double-strand break (DSB), is a serious type of damage that induces genome instability [1]. DSB repair occurs via several pathways to ensure genome integrity [2]. Phosphorylation of histone H2AX is required for all DSB repair pathways to correctly localize DSB repair proteins to the damage sites [3]. When a DSB occurs, serine 139 of H2AX in the vicinity of the DSB is phosphorylated mainly by ATM kinase. Phosphorylated H2AX (γ-H2AX) helps localize DSB repair proteins at the DSB site [3]. Indeed, many DSB repair proteins co-localize with γ-H2AX, and H2AX-deficient cells exhibit chromosomal instability and defects in DSB repair [4]. Another important step in DSB repair is the remodeling of the chromatin structure around DSB sites [5]. In the chromatin remodeling step, the chromatin structure of the DSB site relaxes to allow the recruitment of DSB repair proteins [6]. As described above, precise DSB repair via phosphorylation of H2AX and regulation of the chromatin structure stabilizes the genome and prevents genomic instability-related diseases. DSB repair declines with organismal aging and cellular senescence in mammals [7,8]. However, the molecular mechanisms underlying the abnormalities in DSB repair capacity associated with organismal aging and cellular senescence remain unclear.

Organismal aging occurs when the homeostasis of mammalian bodies declines or collapses with age. Cellular senescence is defined as an irreversible cell cycle arrest characterized by the activation of senescence-associated β-galactosidase (SA β-gal) and the secretory phenotype, called the senescence-associated secretory phenotype (SASP) [9,10,11]. The senescent cell population increases in vivo with organismal aging [12], and there is a positive correlation between the increase in senescent cells and the risk of age-related diseases [13]. There is an obvious link between the symptoms of premature aging and mutations in DNA damage repair, indicating that senescence-associated DNA repair dysfunction and aging-related diseases are linked too [12]. Accumulation of unrepairable DNA DSBs in both telomeric and non-telomeric regions implies that DSB repair function declines with senescence [14]. Elucidation of the molecular mechanisms underlying the decline in DSB repair associated with senescence will help to reduce the risk of aging-related diseases.

Studies on the decline of DSB repair with senescence have shown that in radiation-induced DSB repair, recruitment of DSB repair factors, such as 53BP1, to DSB sites is delayed with organismal aging and cellular senescence [7]. Furthermore, Sedelnikova et al. reported that γ-H2AX focus expansion after irradiation slowed cellular senescence induced by replication stress in vitro [7]. The delayed formation of γ-H2AX that forms early in DSB repair and contributes to accurate and rapid repair is thought to be involved in the overall defects in DSB repair, but the cause of the delayed formation of γ-H2AX in senescence remains unclear.

Senescent cells accumulate profound chromatin structural alterations, leading to the formation of senescence-associated heterochromatin foci (SAHF), characterized by HP1 isoforms, methylated H3K9, and microH2A [10,15]. Chromatin structure is a critical factor for DNA repair [5]. Regulation of HP1 isoforms at DSB site, which are important for heterochromatin formation and maintenance, impacts repair kinetics and pathway selection [16]. Interestingly, in oncogene-induced senescence, SAHFs form around endogenous-persistent DSB sites and suppress the entire DSB response, including H2AX phosphorylation and localization of other repair proteins [17]. However, the relationship between SAHF and the repair of DSBs induced by radiation and other factors remains unclear. Adenosine triphosphate (ATP) is another factor that plays an important role in DSB repair. ATP is not only used for kinase signaling in DSB repair but is also important in the DNA damage response to driving ATP-dependent chromatin remodeling complexes [5,18]. Also, mitochondrial function, which is important for ATP production, is disrupted during cellular senescence [19]. Thus, we investigated the relationship between the delay in γ-H2AX focus formation, SAHF, and ATP.

In this study, γ-H2AX focus formation immediately after irradiation was assessed using early population doubling level (EPDL) versus late population doubling level (LPDL) in primary cells from normal human fetal lung fibroblasts. We also validated γ-H2AX focus formation in human fibroblasts from donors of different ages. In addition, intracellular ATP levels were measured in the EPDL and LPDL. Finally, we investigated the focus formation of γ-H2AX after radiation exposure under SAHF relaxation conditions.

## 2. Results

### 2.1. Delayed Kinetics of γ-H2AX Focus Formation after Irradiation along with Cellular Senescence and Organismal Aging

γ-H2AX is formed early in the DSB repair process and contributes to accurate and rapid repair [3,20,21,22]. However, H2AX phosphorylation is delayed in senescent cells after irradiation [7]. Thus, we investigated whether H2AX phosphorylation in senescent cells was delayed using our experimental system. Cellular senescence of human fetal lung fibroblasts, TIG-3 cells, was induced by cell replication and confirmed using SA β-gal staining (Figure 1A,B). EPDL as non-senescent cells (with relatively low population doubling) and LPDL as senescent cells were irradiated at 0.6 Gy and immunostained for γ-H2AX. Consistent with the previous reports, the number of γ-H2AX foci was lower at 10 min than at 30 min post-irradiation in the LPDL but not in the EPDL (Figure 1C), confirming the delay of γ-H2AX focus formation associated with cellular senescence. Next, we investigated whether the delay in γ-H2AX focus formation, as confirmed in senescent cells, was also observed in organismal aging. Similar to cellular senescence, the number of γ-H2AX foci in human-derived skin cells of different ages, TIG-119 (6 years), TIG-118 (12 years), TIG-103 (69 years), and TIG-107 (81 years), after acute irradiation showed a tendency to increase at 30 min rather than 10 min after irradiation depending on age (Figure 1D). This result indicated that organismal aging delays the formation of γ-H2AX foci.

### 2.2. The Accumulation of DSBs Increases with Cellular Senescence and Organismal Aging under Chronic Low-Dose Irradiation

With the increase in medical exposure, the frequency of radiation exposure to healthy tissues is increasing. Furthermore, considering the increasing life expectancy of modern society and the fact that senescent cells with declining DNA repair capacity increase in vivo with age, the age-dependence of health risks from low-dose long-term exposure should be investigated. Even at unproblematic doses, disruption of the balance between DSB repair and DSB induction due to reduced DSB repair capacity is thought to have a significant hazardous effect on biological systems. Therefore, to evaluate whether DSBs accumulate in senescent cells under chronic low-dose irradiation, cells irradiated at a chronic low dose were stained with the DSB marker γ-H2AX [20]. The EPDL and LPDL cells were irradiated with a dose of 1 Gy (0.694 mGy/min, 1 Gy/day). The number of γ-H2AX foci in LPDL cells immediately after low-dose chronic irradiation was higher than that in EPDL cells (Figure 2A,B). In addition, a comparison of the rate of decrease of the number of γ-H2AX foci between EPDL and LPDL cells 4 h after irradiation showed that DSB repair was slower in LPDL cells (Figure 2C).

DSB repair function also declines with organismal aging [8,23]. Thus, we investigated whether DSBs accumulate under low-dose chronic irradiation in human skin fibroblasts of different ages. The number of γ-H2AX foci tended to be lower in younger (6 and 12 years) cells than in older (69 and 81 years) cells at each time point (Figure 3A,B), even though the population doubling levels (PDLs) of all cells were similar. However, unlike senescent cells, older cells also showed a similar rate of decrease in the number of γ-H2AX immediately after irradiation until 4 h post-irradiation (Figure 3C). This suggests that the decline in the DSB repair function associated with organismal aging is not as severe as that associated with cellular senescence. These results indicate that DSBs are more likely to accumulate under chronic low-dose radiation exposure because of the decline in the DSB repair function associated with cellular senescence and organismal aging.

### 2.3. Delayed γ-H2AX Focus Formation with Cellular Senescence Is Not Related to ATP Levels

Next, to reveal the mechanism underlying the DSB repair function decline in senescent cells, we focused on the relationship between ATP and H2AX phosphorylation. Thus, we analyzed intracellular ATP levels in EPDL and LPDL cells. In the unirradiated condition, the intracellular ATP levels were higher in LPDL cells than in EPDL cells (Figure 4). The amount of mitochondria increases despite abnormal mitochondrial function [24]. Therefore, it is not surprising that ATP levels were elevated in replicative stress-induced senescent cells in this study. Importantly, a significant change in the ATP pool after irradiation was not detected in our experimental setting, and the ATP level 10 min after irradiation was also found to be higher in LPDL cells than in EPDL cells, similar to the unirradiated condition. These data indicate that senescent cells pool more ATP than non-senescent cells, where the amount of ATP to activate the kinase after DSB generation would likely be sufficient, suggesting that there is no correlation between the delay of γ-H2AX focus formation and ATP levels.

### 2.4. Inhibition of SAHF Rescues Delayed γ-H2AX Focus Formation in Senescent Cells

The chromatin structure at DSB sites is a major factor in determining repair kinetics [5,25], and senescent cells are characterized by the formation of SAHF [10,15]. Therefore, we focused on the relationship between SAHF and γ-H2AX foci formation after irradiation. We observed the positional relationship between the γ-H2AX focus and SAHF at 10 min after irradiation and found that most of the γ-H2AX foci were detected outside of SAHF (Figure 5A). In addition, the number of γ-H2AX focus was showed a tendency to decrease in SAHF-positive cells compared to SAHF-negative cells at 10 min after irradiation in LPDL cells (*p* = 0.064) (Figure 5B). This finding suggests that the γ-H2AX focus formation at the DSB site is suppressed in the SAHF regions. Since LiCl has been reported to inhibit SAHF [26], we treated senescent cells with LiCl and evaluated the formation of γ-H2AX foci after radiation exposure. SAHF in LPDL cells was suppressed with 20 mM LiCl for 144 h (Figure 5C,D). The number of γ-H2AX foci in irradiated LiCl-treated LPDL cells was increased at 10 min after irradiation compared with that in untreated LPDL cells (Figure 5E,F). To confirm γ-H2AX focus formation on cells with unsuppressed SAHF despite LiCl treatment, we also compared number of γ-H2AX foci with SAHF suppressed cells and not suppressed in LiCl-treated LPDL cells. The average number of γ-H2AX foci between the SAHF-suppressed cells and the SAHF-unsuppressed cells showed no significant difference (Figure 5G). The data suggest that LiCl treatment increases the rate of γ-H2AX foci formation even in senescent cells that have not completely suppressed the SAHF structure by the effect of LiCl on the SAHF. These data indicate that SAHF is an obstacle to the formation of γ-H2AX foci after irradiation and might be responsible for the DSB repair decline during senescence.

## 3. Discussion

Human life expectancy is increasing, along with the frequency of radiation exposure. Radiation exposure of healthy tissues during radiation therapy for cancer and other medical conditions, such as interventional radiology, is a non-negligible risk of exposure to the human body. To elucidate the relationship between exogenous DSB accumulation and the decline in DSB repair with organismal aging and cellular senescence, we performed chronic irradiation of cells. Here, when 1 Gy irradiation was administered at a dose rate of 1 Gy/day (0.694 mGy/min), the DSB level increased immediately after irradiation with organismal aging and cellular senescence. This indicates that DSBs that occur continuously under chronic irradiation accumulate as repair does not keep pace with the damage. Since the accumulation of DSBs induces cellular senescence, these data indicate that cellular senescence is more likely to be induced by the accumulation of DSBs under chronic low-dose irradiation. In this study, we did not confirm the induction of cellular senescence after chronic irradiation in fibroblasts from younger and older donors; thus, it is necessary to confirm the activation of cellular senescence markers, such as SA β-gal, p21, and p16INK4a. In addition, as shown in Figure 2C, most of the DSBs in senescent cells were not repaired, even 4 h after irradiation. Therefore, it is important to determine age-dependent radiation risk by clarifying the molecular mechanism of critical DSB repair capacity loss associated with cellular senescence. Interestingly, γ-H2AX foci tended to increase 1 h after irradiation rather than immediately after the end of irradiation as shown in Figure 2B and Figure 3B. The detailed mechanism of this phenomenon is still unknown but is consistent with a previous report by Nair et al. [27].

The decline in DSB repair with cellular senescence is caused by multiple factors, including decreased expression of KU proteins and SART6, and delayed localization of 53BP1 to DSB sites [7,28,29]. In this study, we focused on delayed γ-H2AX focus formation as one of the causes of the decline in DSB repair. As H2AX phosphorylation is a key step in DSB repair [21,22,30], proper regulation of H2AX is critical for efficient DSB repair and genomic stability. However, γ-H2AX focus formation after DSB induction is delayed by cellular senescence [7], and we confirmed the delayed kinetics of γ-H2AX in this study. Therefore, to elucidate the cause of the delayed γ-H2AX focus formation, we first focused on ATP levels, which are important for phosphorylation and chromatin remodeling [5,18,25]. Furthermore, abnormal mitochondria are observed during cellular senescence [19,24,31]. Interestingly, intracellular ATP levels were higher in senescent cells than in non-senescent cells, regardless of radiation exposure. These data indicate that ATP is present at a sufficient level for γ-H2AX focus formation on senescent cells and does not affect the deficiency of H2AX phosphorylation. Although mitochondrial membrane potential and respiratory capacity are decreased in senescent cells, the increased mitochondrial mass due to decreased mitophagy might be enough to produce ATP in senescent cells [24,32,33].

It has been reported that the low frequency of γ-H2AX formation is caused by endogenous DNA damage in SAHF regions in senescent cells [17]. In addition, the recruitment of repair proteins is generally slow in DSBs in heterochromatin structures [5,18,25]. Therefore, the γ-H2AX focus formation may be suppressed in the SAHF regions. However, the effects of SAHF after radiation-induced DSBs are unknown. In this study, we clearly showed that the γ-H2AX focus was formed to avoid SAHF immediately after the irradiation of senescent cells. Furthermore, we found that the γ-H2AX focus number immediately after irradiation was higher in senescent cells in which the SAHF structures were suppressed using LiCl. Although SAHF inhibited γ-H2AX focus formation at 10 min after irradiation, there was no difference in the number of γ-H2AX foci between SAHF-positive and SAHF-negative cells in senescent cells at 30 and 60 min. It suggests that SAHF does not completely block γ-H2AX focus formation, it might be possible to slowly relax SAHF at the DSB site to form γ-H2AX focus. Therefore, our data indicate that SAHF inhibits γ-H2AX focus formation and DSB repair within 30 min immediately after DSB in-duction by radiation. As phosphorylation of H2AX is mainly performed by ATM kinase, the results of this study suggest that ATM may not be able to easily access the DSBs generated in the SAHF regions. If SAHF inhibits the recruitment of ATM, an orchestrator that phosphorylates DSB repair proteins [25,30,34], this might explain the decline in DSB repair capacity with cellular senescence. Notably, Di Micco et al. reported that γ-H2AX and ATM foci formation were suppressed in SAHF regions in unirradiated senescent cells [17]. In fact, they confirmed that suppression of SAHF increases γ-H2AX signaling even though DSB is not externally induced and that the enhanced DNA damage response increases apoptosis [17]. ATM activation induced by senescence-related DSBs in non-SAHF regions is involved in the release of SASPs, including inflammation-inducing factors [17,35]. Therefore, SAHF might contribute to sustained SASP factor release by controlling DNA damage signaling, including ATM activation. This study demonstrates that SAHF also suppresses γ-H2AX focus formation by exogenous damage. It is unclear how SAHF affects SASP and apoptosis in senescent cells following induction of exogenous damage, and future studies are required to identify the mechanisms underlying SAHF-regulated effects.

In this study, we found that SAHF produced by fibroblasts, during cellular senescence, inhibits γ-H2AX formation, providing an opportunity to design targeted cancer therapies. Cellular senescence is involved in SASP-induced tumorigenesis promotion and treatment resistance [36]. In senescent cells, it is unclear how SAHF-driven γ-H2AX inhibition, in response to external stimuli, affects SASP and apoptosis induction, and it would be interesting to investigate the impact of SAHF on cancer therapy. In addition, there is phenotypic overlap between senescent fibroblasts and cancer-associated fibroblasts (CAFs) [37]. The CAF are an important component of the tumor microenvironment (TME) [38,39]. The CAF within the TME exhibits a heterogeneous cellular phenotype as well as tumor-promoting and tumor-suppressive roles [38,39]. It is unknown whether CAFs have an SAHF-like phenotype; however, it would be interesting to focus on the epigenetics and DNA damage response to the CAF. In addition, mitochondrial metabolism is linked to cellular senescence, CAF, and tumors [40,41,42]. Mitochondria removal from senescent cells suppresses the development of cellular senescence-associated phenotypes, including SASP and SAHF [42]. MitoTam, an agent that targets mitochondria to induce cellular apoptosis, causes senescent cell elimination, both in vitro and in vivo [41]. These findings indicate that mitochondria play a key role in cellular senescence. In addition, CAFs promote mitochondrial activity in cancer [39]. In this study, we found that SAHF inhibits γ-H2AX formation in response to irradiation; this finding improves the understanding of the biological responses induced by the combination of radiation therapy and mitochondria-targeted therapeutic agents.

## 4. Materials and Methods

### 4.1. Cell Culture

The human fetal lung fibroblast cell line TIG-3 and human skin fibroblast cells, i.e., TIG-119 (6 years old, male), TIG-118 (12 years old, female), TIG-103 (69 years old, female), and TIG-107 (81 years old, female), were purchased from JCBR Cell Bank, National Institutes of Biomedical Innovation, Health, and Nutrition, Osaka, Japan. The cells were cultured in Dulbecco’s modified Eagle’s medium (043-30085; FUJIFILM Wako, Osaka, Japan) containing 10% fetal bovine serum (173012; Sigma-Aldrich, St. Louis, MO, USA) and 1× penicillin and streptomycin (168-23191; FUJIFILM Wako, Osaka, Japan) at 37 °C in a humidified atmosphere with 5% CO_2_. TIG-3 cells with PDL of 30–45 were used as EPDL cells and those with PDL > 55 as LPDL cells. TIG-119 and TIG-103 cells were used at PDL between 24 and 40, respectively. TIG-118 and TIG-107 cells were used between PDL of 24 and 29, respectively.

### 4.2. Senescence-Associated β-Galactosidase

The cultured cells were fixed with 4% formaldehyde (163-20145; FUJIFILM Wako) for 3 min at approximately 23 °C. The cells were washed with 1× PBS and incubated with SA β-gal staining solution (0.02 M citric acid, 0.04 M sodium phosphate buffer, 5 mM potassium ferricyanide, 5 mM potassium ferrocyanide, 150 mM sodium chloride, 2 mM magnesium chloride, and 1 mg/mL 5-bromo-4-chloro-3-indolyl-β-D-galactoside; pH 6.0) for 12 h at 37 °C. LPDL cells were confirmed using SA-β-gal staining to induce cellular senescence in each experiment.

### 4.3. Radiation Exposure

A γ-ray irradiation device with cesium-137 as the radioisotope source was used for chronic low-dose irradiation experiments at a dose of 1 Gy (0.694 mGy/min, 1 Gy/day) at 37 °C in a humidified atmosphere with 5% CO_2_ at the Radiation Biology Center, Kyoto University. An X-ray irradiation device (OM-B205; Omic Corporation, Ritto, Japan) was used for observing the formation of γ-H2AX focus under an irradiation dose of 0.6 Gy (0.5 Gy/min) at approximate 23 °C.

### 4.4. Immunofluorescence Staining

Cells were fixed in 4% paraformaldehyde for 10 min at approximately 23 °C and permeabilized overnight with 70% ethanol at 4 °C. The cells were blocked with 4% bovine serum albumin, 0.1% Triton X-100, and 0.5% Tween 20 in 1× PBS for 1 h at approximate 23 °C after hydrophilic treatment with 1× PBS for 30 min at approximately 23 °C. The cells were then stained with antibodies against γ-H2AX (1:1000; 05-636; Millipore, MA, USA) and tri-methyl-H3K9 (1:500; ab8898; Abcam, Cambridge, UK). Secondary antibodies conjugated with Alexa488 or Alexa555 were added and incubated for 1 h at approximately 23 °C. VECTASHIELD^®^ Antifade Mounting Medium with propidium iodide or with DAPI (Vector Laboratories, Newark, CA, USA) was used as the mounting agent. Fluorescence was detected using a fluorescence microscope (BX53; Olympus, Tokyo, Japan).

### 4.5. ATP Level Assay

To measure ATP levels, we used ATPliteTM (6016943; Perkin Elmer, Waltham, MA, USA), an intracellular ATP level assay kit, according to the manufacturer’s instructions. Briefly, the cells were cultured in 96-well plates (Thermo Fisher Scientific, Waltham, MA, USA) and incubated at 37 °C and 5% CO_2_ for 24 h. After incubation, the culture medium was aspirated, and 100 μL of fresh serum-free DMEM was added to each well. The cells were irradiated with X-rays at 5 Gy. After 10, 30, or 60 min of incubation, 50 μL of mammalian cell lysis solution was added to each well, and the 96-well plates were shaken at 700 rpm for 5 min on a shaking machine (Eppendorf, Hamburg, Germany). Then, 50 μL of the substrate solution was added to each well, and the plates were shaken at 700 rpm for 5 min. After allowing the plates to stand in the dark for at least 10 min, ATP levels were calculated by detecting the amount of luminescence using MicroBeta2 (Pekin Elmer, Waltham, MA, USA).

### 4.6. Lithium Chloride Treatment

LiCl, a SAHF inhibitor, was used to relax SAHF in senescent cells. LiCl (121-05242; FUJIFILM Wako), purchased in powder form, was prepared to a final concentration of 20 mM using the culture medium as a solvent. The LiCl solution was filtered and sterilized using a syringe filter (SLPES2522S; Hawach Scientific, Xi’an, China) with a pore size of 0.22 μm.

### 4.7. Statistical Analysis

Differences in the number of γ-H2AX foci and the rate of SAHF-positive cells between the sample groups were analyzed using Student’s *t*-test at each time point using Microsoft Excel 2016. *p* < 0.05 was considered statistically significant.

## 5. Conclusions

Our results clearly show accumulation of DSBs is caused by the senescence-dependent decline in DNA damage repair under chronic low-dose radiation exposure. Second, our data indicate that SAHF inhibited radiation-induced DSB repair and delayed γ-H2AX focus formation, which is one of the factors responsible for the decline in DSB repair capacity. Pharmacological inhibition of SAHF rescued the rate of γ-H2AX focus formation. This suggests that SAHF may be a therapeutic target for the resolution of cellular senescence-dependent DSB-repair dysfunction.

## Figures and Tables

**Figure 1 ijms-25-03355-f001:**
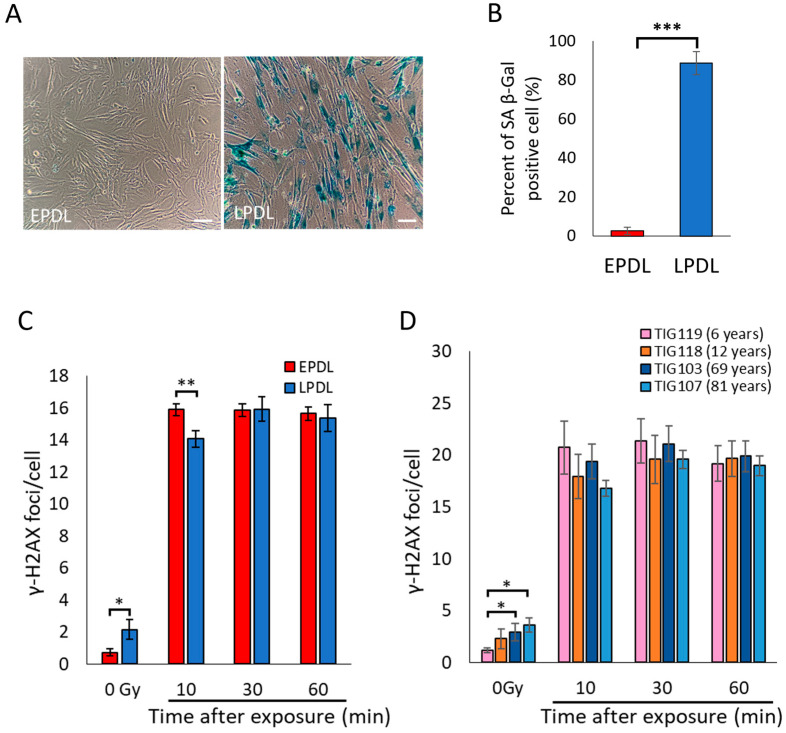
γ-H2AX focus formation delays with cellular senescence and organismal aging. (**A**) Confirmation of cellular senescence induction in TIG-3 cells (Early population doubling level: EPDL, Late population doubling level: LPDL) using senescence-associated β-galactosidase (SA β-gal staining). Images were taken at 100× magnification. Scale bars: 100 μm. (**B**) The percent of SA β-gal positive cell (blue stained) were calculated by counting. Error bars indicate standard error. *t*-test: *** *p* < 0.001, *n* = 4. (**C**) The TIG-3 cells were immunostained for γ-H2AX after 0.6 Gy (0.5 Gy/min) of X-ray exposure. The average number of γ-H2AX foci per cell was obtained by counting in 100 cells. Error bars indicate standard error of the mean (SEM). *t*-test: * *p* < 0.05, ** *p* < 0.01, 0 Gy and 10 min: *n* = 7, 30 min and 60 min: *n* = 5. (**D**) TIG-119, TIG118, TIG103, and TIG-107 cells after 0.6 Gy (0.5 Gy/min) of X-ray exposure were immunostained for γ-H2AX. The average number of γ-H2AX foci per cell was obtained by counting in 100 cells. Error bars indicate SEM. *t*-test: * *p* < 0.05, TIG119 and TIG103: *n* = 7, TIG118 and TIG107: *n* = 5.

**Figure 2 ijms-25-03355-f002:**
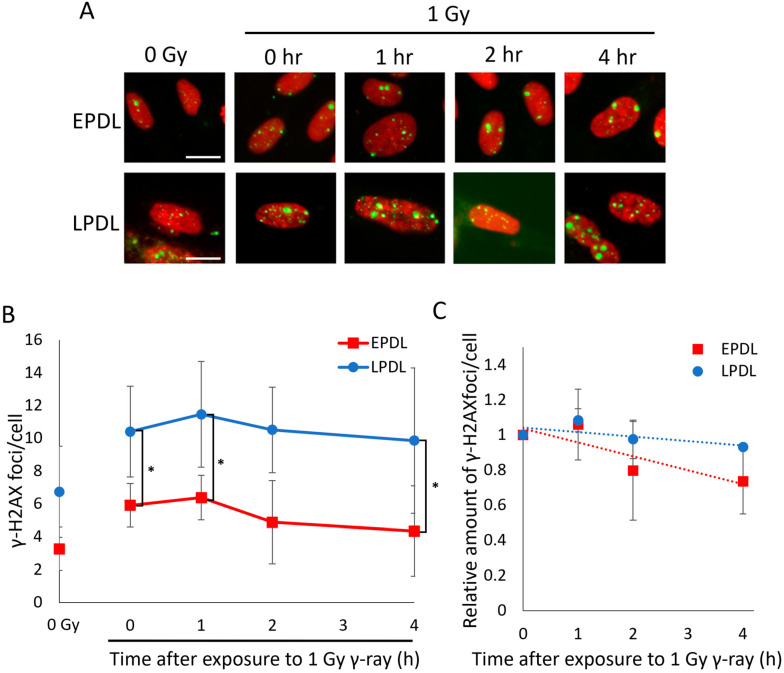
Low-dose chronic irradiation-induced DNA damage accumulation in senescent cells. (**A**) Early population doubling level (EPDL) and late population doubling level (LPDL) cells were fixed immediately (0 h) and 4 h after 1 Gy (0.694 mGy/min, 1 Gy/day) γ-irradiation and immunostained for γ-H2AX. The figure shows the representative images of immunostaining for γ-H2AX (Alexa 488: green) and the nucleus (propidium iodide: red). Images were taken at 400× magnification. Scale bars: 20 μm. (**B**) The average number of γ-H2AX foci per cell was obtained by counting in 100 cells. (**C**) The number of γ-H2AX foci at each time point relative to the number of γ-H2AX foci immediately after irradiation. Error bars indicate SD. *t*-test: * *p* < 0.05, EPDL: *n* = 3, LPDL: *n* = 4.

**Figure 3 ijms-25-03355-f003:**
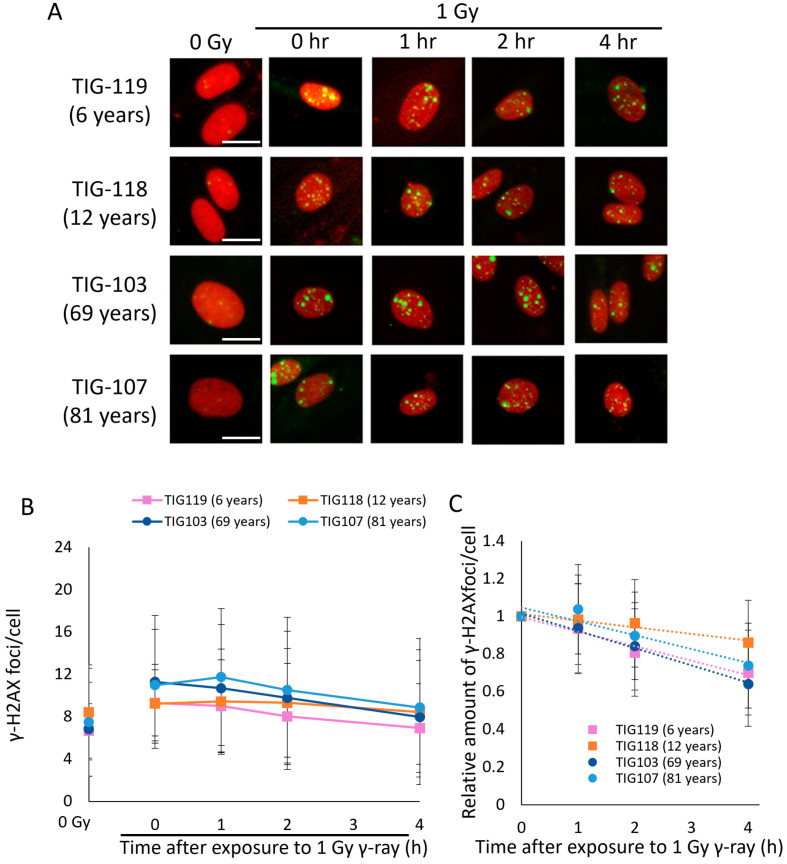
Accumulation of DSBs with age under low-dose chronic irradiation. (**A**) TIG-119, TIG118, TIG-103, and TIG-107 cells were fixed immediately (0 h) and 4 h after 1 Gy (0.694 mGy/min, 1 Gy/day) γ-irradiation and immunostained for γ-H2AX. Representative images show immunostaining for γ-H2AX (Alexa 488: green) and the nucleus (propidium iodide: red). Images were taken at 400× magnification. Scale bars: 20 μm. (**B**) The average number of γ-H2AX foci per cell was obtained by counting in 100 cells. (**C**) The number of γ-H2AX foci at each time point relative to the number of γ-H2AX foci immediately after irradiation. Error bars indicate SD. *n* = 3.

**Figure 4 ijms-25-03355-f004:**
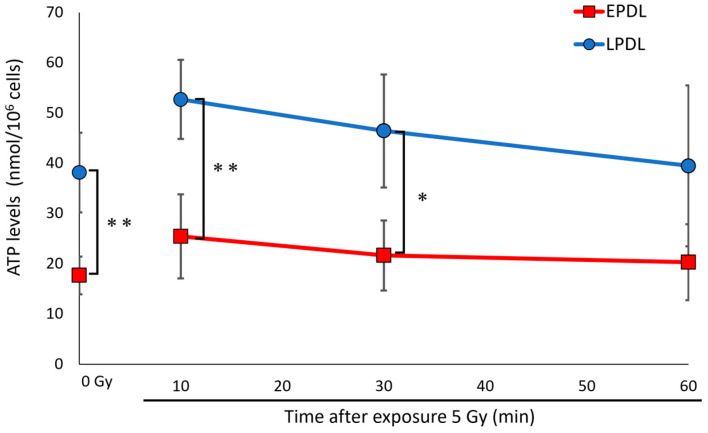
The delay of γ-H2AX focus formation with cellular senescence is independent of the Adenosine triphosphate (ATP) levels. Early population doubling level (EPDL) and late population doubling level (LPDL) cells were irradiated with 5 Gy (4 Gy/min) X-rays. ATP levels were measured using ATPliteTM before and 10, 30, and 60 min after irradiation. ATP levels per 1 × 10^6^ cells were calculated. Error bars indicate SD. *t*-test: * *p* < 0.05, ** *p* < 0.01, *n* = 3.

**Figure 5 ijms-25-03355-f005:**
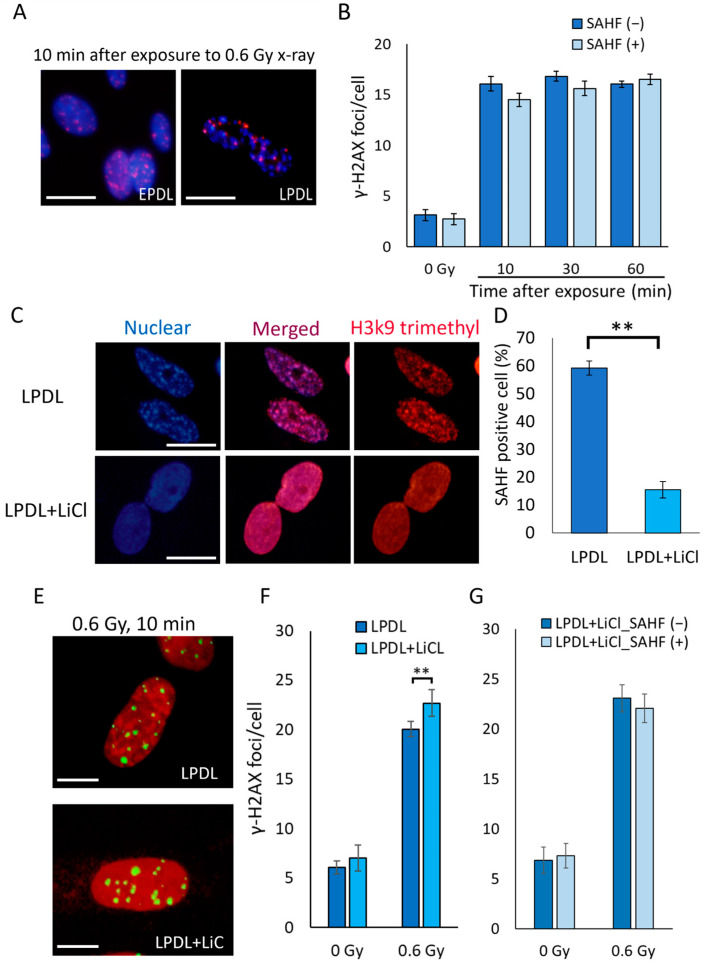
SAHF disrupts the γ-H2AX focus formation. (**A**) Early population doubling level (EPDL) and late population doubling level (LPDL) cells were fixed 10 min after 0.6 Gy (0.5 Gy/min) X-ray irradiation and immunostained for γ-H2AX (Alexa 555: red) and the nucleus (DAPI: Blue). Images were taken at 600× magnification. Scale bars: 20 μm. (**B**) The average number of γ-H2AX foci per cell was obtained separately for senescence-associated heterochromatin foci (SAHF) negative cells (SAHF (−)) and SAHF positive cells (SAHF (+)) in LPDL cells. Error bars indicate standard error of the mean (SEM). 0 Gy and 10 min: *n* = 6, 30 min and 60 min: *n* = 5. (**C**) LiCl-treated (LPDL + LiCl) and untreated LPDL cells were immunostained for trimethyl H3K9 as a marker of heterochromatin (Alexa 555: red) and the nucleus (DAPI: Blue). Images were taken at 400× magnification. Scale bars: 20 μm. (**D**) More than 100 cells were counted to obtain the relative value of SAHF-positive cells. Error bars indicate SD. *t*-test: ** *p* < 0.01, *n* = 4. (**E**) LiCl-treated (LPDL + LiCl) and untreated LPDL cells were fixed 10 min after 0.6 Gy (0.5 Gy/min) of X-ray irradiation and immunostained for γ-H2AX. Representative images show immunostaining for γ-H2AX (Alexa 488: green) and the nucleus (propidium iodide: red) after X-ray exposure. Images were taken at 400× magnification. Scale bars: 10 μm. (**F**) The average number of γ-H2AX foci per cell was obtained by counting in 100 cells. Error bars indicate SEM. *t*-test: ** *p* < 0.01, *n* = 6. (**G**) In LiCl-treated LPDL cells, the average number of γ-H2AX foci per cell was determined separately for SAHF-suppressed cells (LPDL + LiCl_SAHF (−)) and not suppressed cells (LPDL + LiCl_SAHF (+)). Error bars indicate SEM, *n* = 6.

## Data Availability

The data presented in this study are available on request from the corresponding author.

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
