# Peer review of "Senescence-Associated Heterochromatin Foci Suppress γ-H2AX Focus Formation Induced by Radiation Exposure"

_ijms, 2024, doi:10.3390/ijms25063355_

Round 1

Reviewer 1 Report

Comments and Suggestions for Authors

The study investigates the effect of cellular senescence and aging on DNA damage repair and DSB, particularly focusing on the role of senescence-associated heterochromatin foci (SAHF) in suppressing γ-H2AX focus formation after radiation exposure. The authors suggest that senescence delays the formation of γ-H2AX foci, a key step in DNA double-strand break repair. The study suggests that the presence of SAHF in senescent cells interferes with this process, potentially leading to accumulated DNA damage. The study, while insightful, has several noted concerns regarding its methodology and data interpretation.

•            The authors considered PDL 30-45 as “early” PDL, and PDL>55 as “late PDL”. However, these PDLs may not be different enough to allow a clear distinction between the behavior of proliferating and senescent cells. Typically, with fibroblasts like IMR-90 or WI-38 cells, early PDL are considered between 20-30. Therefore, younger proliferating cells may be useful to allow a clear distinction in the foci formation between early and late PDLs.

•            Figure 1C and Figure 5F,G: the differences do not look convincing enough, nor biologically significant. I) Assessing a major number of cells/nuclei, and (II) a better experimental design with appropriate PDLs, could be beneficial to see clear differences.

•            The authors suggests that, in senescent cells, the formation of y-H2AX following DNA damage is delayed, even though the differences do not look significant (Figure 1). It is well-known, however, that senescent cells have a high level of y-H2AX compared to proliferating cells, indeed y-H2AX is also used as a non-specific marker of cellular senescence. Further, the authors observe the accumulation of H2AX in senescent cells compared to proliferating cells (Figure 2). How do the authors explain this? Is there a delayed induction of H2AX, but also a lower resolution of DSB?

•            Figure 3B: would it be helpful to monitor the formation of foci for longer time after the 4 hr irradiation?

Reviewer 2 Report

Comments and Suggestions for Authors

The IR induced  gamma H2AX is questionable. The frequency of foci per cell is very low as compared to the reported literature. The quality of the figure showing foci is not acceptable. No scale for the cells is shown. The residual gamma H2AX is questionable. Significant difference in the foci reduction is not significant. No clear information is given about the initial and residual foci, which makes it difficult to correlate with the SAHF.

The authors may consider to discuss the important published literature.

Comments on the Quality of English Language

The authors need to discuss in simple language the differences in chromatin and its correlation with DSB repair as described in past. They need to discuss the role of HP1 isoforms in DSB repair.

Reviewer 3 Report

Comments and Suggestions for Authors
Summary: an in vitro study to explore the mechanism underlying the inhibition of DNA damage repair in senescence. 

Major comment:

1. It is suggested the authors add the potential applications of this finding in a separate section in the Discussion section. I suggest the following ones in case of cancer:

1-1. This study on fibroblasts demonstrated that senescence-associated heterochromatin foci (SAHF) can mediate the senescence process in fibroblasts through inhibition of γ-H2AX formation. Recent studies have shown that senescent fibroblast cells exhibit cancer-associated fibroblast (CAF) phenotype (https://www.ncbi.nlm.nih.gov/pmc/articles/PMC9498467/). This issue can improve cancer progression (https://pubmed.ncbi.nlm.nih.gov/37784082/) and treatment resistance (https://pubmed.ncbi.nlm.nih.gov/38032584/).  The findings of this study can provide an opportunity to design targeted therapies against CAF formation. to improve the treatment results. 

1-2. Recent reports have delineated the crucial impact of mitochondrial metabolism in cancer progression and treatment resistance (https://pubmed.ncbi.nlm.nih.gov/37627086/). In 2018, Hubackova et al. demonstrated that targeting mitochondria (using MitoTAM) can eliminate senescent cells (https://www.nature.com/articles/s41418-018-0118-3). This might reflect the crucial role of mitochondria in the senescence process. In support, it has been shown that mitochondria play a central role in SAHF formation (https://www.ncbi.nlm.nih.gov/pmc/articles/PMC5053570/). The findings of this study can provide an opportunity to design targeted therapies to improve treatment response.

It is suggested the authors mention this information, using all the noted references., to expand the clinical implications of this study and improve the bibliography.  

Minor comments:

2. Keywords: To improve the tracking of the paper, It is suggested to use the MeSH term. SAHF can be substituted with its complete form.

3. Regarding the information presented in the Introduction section, it is suggested the authors cite ONE most relevant reference.  This can enable to tracking of the presented information. 

4. Figure captions: It is suggested to explain the full term of abbreviations applied in the figures. 

5. Discussion: it is suggested the Discussion section starts with a practical summary of the study findings, continues with a comparison of the study findings with the corresponding literature, noting the potential applications of the findings,  and ends with study limitations, strengths, and suggestions for future works.  

6. Statistical analysis: please add the significance value (e.g., p-value <0.05 was considered significant). “

Round 2

Reviewer 1 Report

Comments and Suggestions for Authors

The differences in several experiments (Figure 1C,1D, 5B,5F,5G) are not convincing enough and it is not sufficient to remove the word "significant" from the text. I would encourage the authors to increase the sample size and see if there is a stronger significance.

Reviewer 2 Report

Comments and Suggestions for Authors

After carefully reading this article, the information provided in this review article does not advance the field of DDR.

Comments on the Quality of English Language

This information is of very limited used.

Round 3

Reviewer 2 Report

Comments and Suggestions for Authors

I recommend the revised paper for the publication.